# Exposing the hidden influence of selection rules on phonon–phonon scattering by pressure and temperature tuning

Navaneetha K. Ravichandran [1✉] & David Broido [2]

Selection rules act to restrict the intrinsic anharmonic interactions between phonons in all crystals. Yet their influence on phonon propagation is hidden in most materials and so, hard to interrogate experimentally. Using ab initio calculations, we show that the otherwise invisible impact of selection rules on three-phonon scattering can be exposed through anomalous signatures in the pressure ($P$) and temperature ($T$) dependence of the thermal conductivities, $\kappa$, of certain compounds. Boron phosphide reveals such underlying behavior through an exceptionally sharp initial rise in $\kappa$ with increasing $P$, which may be the steepest of any material, and also a peak and decrease in $\kappa$ at high $P$. These features are in stark contrast to the measured behavior for many solids, and they occur at experimentally accessible conditions. These findings give a deep understanding of phonon lifetimes and heat conduction in solids, and motivate experimental efforts to observe the predicted behavior.

[1] Department of Mechanical Engineering, Indian Institute of Science, Bangalore, Karnataka, India. [2] Department of Physics, Boston College, Chestnut Hill, MA, USA. ✉email: navaneeth@iisc.ac.in

The thermal conductivity, $\kappa$, is a fundamental transport parameter that governs the efficiency of heat conduction through solids. In insulating crystals, heat is conducted by phonons, and intrinsic thermal resistance comes from phonon–phonon scattering processes, which arise from the anharmonicity of the interatomic bonding potential[1–3]. Lowest-order processes involving the mutual interaction among three phonons typically dominate this intrinsic resistance.

For a given phonon mode, the collection of all such scattering processes that conserve energy and momentum––the scattering phase space––is dictated entirely by a material's phonon dispersions. Thus, for a phonon with wave vector, $\mathbf{q}$, and polarization, $j$, and phonon frequency, $\omega_{\mathbf{q}j}$, the phase space of energy and momentum conserving three-phonon scattering corresponds to all processes satisfying[2–6]: $\omega_{\mathbf{q}j} \pm \omega_{\mathbf{q}'j'} = \omega_{\mathbf{q}''j''}$ and $\mathbf{q} \pm \mathbf{q}' = \mathbf{q}'' + \mathbf{G}$, where $\mathbf{G}$ is a reciprocal lattice vector. Specific features in these dispersions activate selection rules that can severely restrict the phase space of certain three–phonon scattering channels[2–5,7–13]. Though not arising from underlying symmetries as is the case for other selection rules, the phase space selection rules have a similar effect and can lead to anomalously long phonon lifetimes.

The most well-known of these selection rules dictates that an acoustic phonon cannot decay anharmonically into a set of others that are in the same dispersive phonon branch[7–11]. It explains the intriguing experimental observations of long lifetimes of the large wave vector transverse acoustic (TA) phonons at low temperatures[14,15]. Exceptions exist[16], but the phase space for those scattering events is small, which we have verified from the small phase space found from numerical calculations for many compounds.

The influence of this selection rule on $\kappa$ was long thought to be negligible, since intrinsic thermal resistance is dominated by scattering processes involving phonons in different branches, particularly at higher temperatures. However, it was recently pointed out that, even around and above room temperature (RT; 300 K), this selection rule can be activated in materials whose three acoustic branches come close to each other in some high frequency region of the Brillouin zone[4,5,12]. Acoustic phonons in such a region then see a severely restricted phase space available for decay into two others (AAA processes). This opens the opportunity for achieving long phonon lifetimes and correspondingly large contributions to $\kappa$.

But what about other phonon–phonon scattering events: shouldn't their presence mask any reduction in AAA scattering induced by the selection rule? Indeed, this is the case in almost all materials. However, nature does give at least one exception that shows a profound and observable impact of the AAA selection rule on $\kappa$: cubic boron arsenide (BAs). First principles calculations[4] predicted an unusually weak AAA scattering in BAs, resulting from activation of the AAA selection rule. Other features in the BAs phonon dispersions open a frequency window in which only the weak AAA scattering occurs[4,5,12,13]. While inclusion of higher-order four-phonon scattering was found to reduce the BAs $\kappa$[17,18], a RT value of around 1300 Wm$^{-1}$K$^{-1}$ was predicted, by far the highest $\kappa$ of any semiconductor, and behind only diamond and graphite among naturally occurring materials. These theoretical findings have been confirmed by measurements[18–20], and recent theoretical predictions suggest similar behavior in semimetallic TaN[21], adding strong support to the impact of selection rules and corresponding reductions in the three-phonon scattering phase space on $\kappa$.

Recently, we have identified several other compounds, such as cubic boron phosphide (BP) and silicon carbide (3C-SiC), where the AAA selection rule is activated, giving anomalously weak AAA scattering channels[5]. However, unlike in BAs, for these compounds, the weak AAA scattering is masked by another strong three-phonon scattering channel involving two acoustic phonons and an optic (O) phonon (AAO scattering channel), thus drastically reducing its impact on $\kappa$. In this work, we show that the otherwise-hidden weak AAA scattering in BP can be exposed by hydrostatic pressure ($P$) and temperature ($T$) tuning of $\kappa$. We demonstrate that the evolving interplay between AAA and AAO scattering channels with varying $P$ and $T$ gives rise to a peak and subsequent drop in $\kappa$ at high $P$. Furthermore, in BP, an unusually sharp rise in $\kappa$ occurs as $P$ increases from ambient pressure, which may be the steepest in any material. These features are contrary to the measured behaviors of $\kappa$ in many other insulating crystals, that instead show a roughly linear increase with $P$ far away from phase transitions[22–29], consistent with simple theories[27,30].

The $\kappa(P, T)$ of BP and 3C-SiC are calculated using a recently developed unified ab initio theoretical framework, which combines the use of density functional theory to obtain phonon dispersions and phonon scattering rates, with a numerical solution of the phonon Boltzmann transport equation[31]. The theory has no adjustable parameters, and it has demonstrated good agreement with the measured $\kappa$ of several materials over a broad temperature and pressure range[5,18,31–33], including BP and 3C-SiC. Both three-phonon and higher-order four-phonon scattering processes are included in the calculations along with phonon scattering by mass disorder arising from the natural isotope mixture on the constituent atoms (naturally occuring boron is 19.9% $^{10}$B, 80.1% $^{11}$B, and phosphorus is 100% $^{31}$P). The details of this first principles approach have been published in refs. [5,31,32] and are summarized in the Methods section below.

## Results

We focus on our results for BP, which show the most striking anomalous behavior of $\kappa(P, T)$ directly tied to the effect of the AAA selection rule. Figure 1 shows the calculated RT $\kappa(P)$ of BP, with and without four-phonon scattering, for three cases: BP with (1) naturally occurring B isotope mix ($^{Nat}$BP), (2) B atoms isotopically enriched to 100% $^{11}$B ($^{11}$BP) and (3) B atoms isotopically enriched to 100% $^{10}$B ($^{10}$BP). Measured results for these three cases at ambient pressure[33,34] are in excellent agreement with the calculations, as shown in Fig. 1. All curves peak at around 55–60 GPa and subsequently decrease with increasing $P$, contrary to the measured linearly increasing behavior of $\kappa(P)$ that

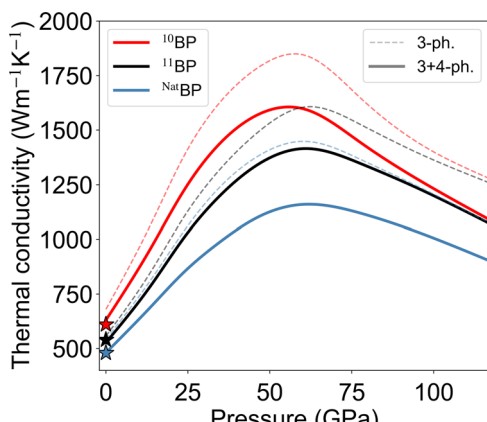

**Fig. 1 Pressure-dependent thermal conductivity of BP.** Room temperature thermal conductivity of BP with naturally occurring B isotope concentration (blue), isotopically pure $^{11}$B (black), and isotopically pure $^{10}$B (red) as a function of hydrostatic pressure, $P$, with (solid) and without (dashed) four-phonon scattering. With increasing $P$, all curves show a sharp rise followed by a peak and subsequent drop, contrary to behavior in many other materials. Stars indicate measured values at ambient conditions[33,34].

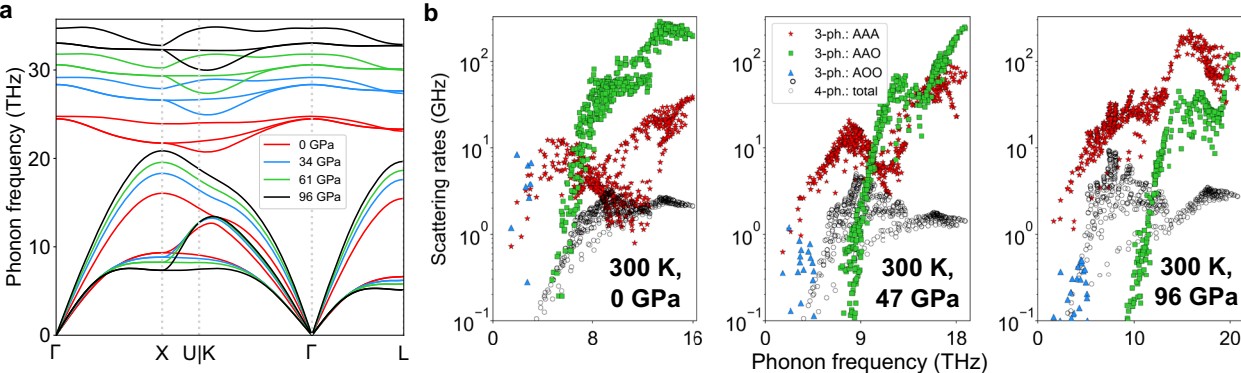

**Fig. 2 Phonon dispersions and phonon–phonon scattering rates of BP vs. pressure. a** Phonon dispersions of BP at different values of *P*. **b** Process-wise three-phonon and total four-phonon scattering rates for BP at RT and different pressures. Since acoustic phonons carry the majority of the heat, only their frequency bandwidth is plotted.

typically occurs in other materials. In addition, the $\kappa(P)$ curves rise rapidly with increasing *P*, and there is a significant increase in the slopes, $d\kappa/dP$, going from [11]BP to [10]BP. The peak values of $\kappa$ are large, around 1150 Wm$^{-1}$K$^{-1}$ ([Nat]BP), 1400 Wm$^{-1}$K$^{-1}$ ([11]BP), and 1600 Wm$^{-1}$K$^{-1}$ ([10]BP), about 2.5 times higher than their values at ambient pressure. It is important to note that, the effect of four-phonon scattering on $\kappa$ is relatively weak at all pressures in BP. Thus, the non-monotonic $\kappa(P)$ behavior for BP is unrelated to that predicted for BAs[32], as discussed below.

To understand these pressure-dependencies of $\kappa$, we first examine the phonon–phonon scattering processes in BP at ambient pressure. The light atoms and stiff bonding of BP result in a high $\kappa$ of around 500–600 Wm$^{-1}$K$^{-1}$ at RT and ambient pressure[33–35]. BP crystallizes in the zinc blende structure and has two atoms in each unit cell. A picture of the crystal structure is given in Supplementary Fig. 1. The phonon dispersions then consist of three A and three O phonon branches. Only three types of three-phonon scattering processes can occur, which involve combinations of A and O phonons: AAA, AAO, and AOO. Energy conservation forbids all other processes (e.g., OOO)[5]. Figure 2b shows the RT AAA, AAO and AOO scattering rates for BP at various pressures, along with those for four-phonon scattering. The sharp dip in the AAA scattering rates at $P = 0$ in Fig. 2b results from activation of the AAA selection rule driven by the close proximity of the three acoustic phonon branches along the $\Gamma \rightarrow K$ direction, as seen in Fig. 2a. Similar dips driven by the same selection rule occur in other cubic compounds such as BAs, BSb, 3C-SiC, diamond, c-BN, c-GaN, and c-AlN[5,12] as well as in certain transition metal carbides[36].

Apart from the close proximity of the acoustic phonons, other features in the phonon dispersions activate additional selection rules. The mass differences between constituent atoms of these compound semiconductors cause a frequency gap, $\omega_{\text{A-O}}$, between acoustic and optic phonons (A–O gap). Energy conservation mandates that only those acoustic phonons with frequencies greater than $\omega_{\text{A-O}}$ can participate in AAO scattering processes[4]. Similarly, energy conservation restricts the participation of acoustic phonons in AOO scattering processes to those with frequencies less than the optic phonon bandwidth, $\omega_{\text{O}}$[5,13]. These restrictions are illustrated in Fig. 2 of ref. [5] and Supplementary Fig. 3.

The relatively small phosphorus to boron mass ratio ($M_P/M_B = 2.6$) gives an onset of AAO scattering for acoustic phonons ($\omega_{\text{A-O}}$) at around 6 THz, which nearly coincides with the onset of the AAA dip, as discussed below. As shown in Fig. 2b, the AAO scattering rates in BP dominate over almost the entire frequency region of the dip in AAA scattering rates at ambient pressure and

RT. This behavior is in stark contrast to large mass ratio compounds like BAs ($M_{\text{As}}/M_B = 7$), where $\omega_{\text{A-O}}$ is large enough to almost completely remove AAO processes for acoustic phonons, thereby fully exposing the sharp dip in the AAA scattering rates, and resulting in significantly enhanced three-phonon limited $\kappa$[4]. Thus, the isoelectronic substitution of As with P activates strong AAO scattering that hides the otherwise similar behavior BP has, to BAs. A comparison of room temperature three-phonon and four-phonon scattering rates in BP and BAs is given in Supplementary Fig. 8. To illustrate the importance of the AAO processes in BP, we have performed calculations of $\kappa$ in which they have been artificially turned off. These calculations show that (i) the three-phonon limited $\kappa$ of BP then increases by over an order of magnitude, exceeding 6000 Wm$^{-1}$K$^{-1}$, and (ii) inclusion of four-phonon scattering dramatically reduces the $\kappa$ of BP to around 2600 Wm$^{-1}$K$^{-1}$. This ultrahigh thermal conductivity and its sharp reduction from four-phonon scattering thus obtained for BP are qualitatively similar to the behavior in BAs. Note that, in BP and BAs, the optical phonon bandwidth is small, so the AOO scattering channel only affects low frequency phonons well below the AAA dip region; so it is not important for the discussed behavior.

In BP, application of hydrostatic pressure introduces three intertwined behaviors that affect $\kappa$: (i) the longitudinal acoustic (LA) and optic phonons shift to higher frequencies [Fig. 2a], which increases LA phonon group velocities and weakens the AAO scattering rates due to decreased optic phonon occupations. Both of these changes contribute to the increasing $\kappa$ with *P* found in many materials. Two additional features are critical to explain the anomalous behavior seen in Fig. 1 - (ii) optic phonons stiffen faster than do acoustic phonons [see Fig. 2a], which increases $\omega_{\text{A-O}}$ (see Supplementary Fig. 4) and shifts the onset of AAO processes to higher frequencies, thereby exposing increasingly large portions of the AAA dip. This behavior increases the rate of rise in the $\kappa(P)$ of BP in the low *P* range, as discussed below; and (iii) with increasing *P*, the LA phonons stiffen, while the transverse acoustic (TA) phonons weakly soften. The resulting increased separation between LA and TA phonon branches gradually removes the impact of the AAA selection rule, and so increases the AAA scattering rates relative to those at ambient pressure [see Fig. 2b]. This behavior acts to drive $\kappa$ lower, as pointed out previously[32,37].

**Peak and decrease in $\kappa$.** At RT and low *P*, the influence of trends (i) and (ii) listed above is stronger than trend (iii) resulting in an increasing $\kappa(P)$. Above 50 GPa, the continued shift of AAO processes towards higher phonon frequencies almost fully

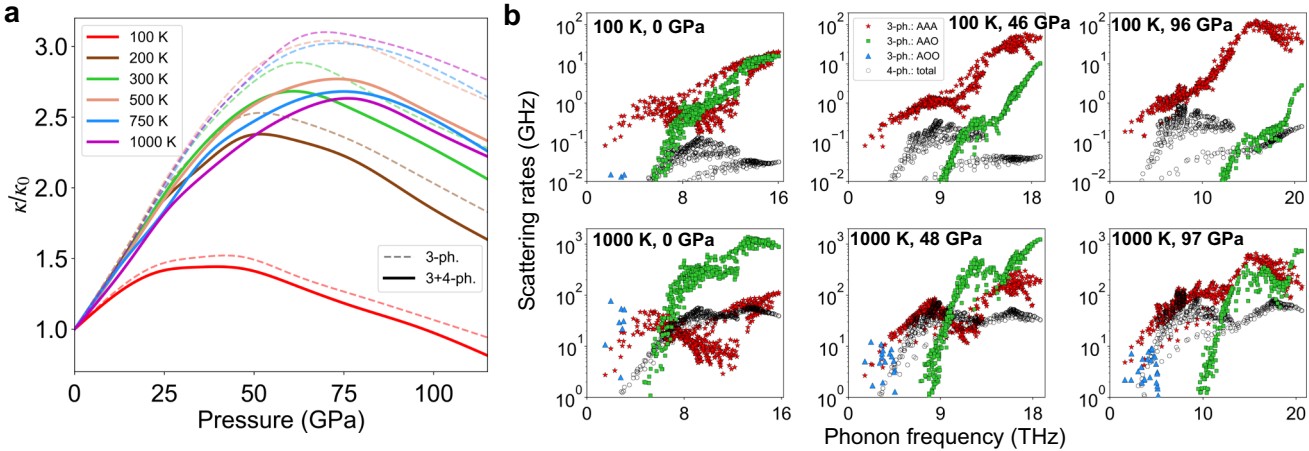

**Fig. 3 Temperature dependence of the scaled thermal conductivity and phonon–phonon scattering rates of BP vs. pressure. a** Pressure dependence of the $\kappa$ of BP scaled by its zero pressure values, $\kappa_0$, for different temperatures. **b** Process-wise three-phonon and total four-phonon scattering rates for BP at 100 K and 1000 K, and at different pressures.

exposes the AAA dip, and the rising AAA scattering rates from trend (iii) eventually dominate the behavior, causing $\kappa$ to decrease with increasing $P$. As a result of this evolving interplay between AAA and AAO processes, the RT $\kappa(P)$ achieves a peak value at around 60 GPa, which is around 2.5 times that of its value at $P = 0$.

Figure 3a shows the $\kappa(P)$ for BP scaled by its zero pressure value, $\kappa_0$, for different $T$. A striking difference in the $T$-dependences of $\kappa(P)$ below RT is seen when compared with those around and above RT. Around and above RT, the curves for each $T$ roughly overlap, indicating that the $P$ and $T$ dependencies are separable. Such separability is also predicted from empirical theory[26,27] and is found in many compounds[22–29] such as MgO. That it also occurs in BP with its anomalous non-monotonic $\kappa(P)$, is a consequence of the independent changes in AAA and AAO scattering rates induced by changing $T$ or $P$, shown in Fig. 3b for $T = 100$ K and $T = 1000$ K. At 100 K, the AAO scattering rates simultaneously shift to higher frequencies and weaken rapidly with increasing pressure. However, around and above 300 K, their shift to higher frequencies occurs at the same rate as at 100 K, but the weakening of the magnitudes is minimal. These differences are not caused by changes in phonon dispersions with temperature. Indeed, the phonons in this strongly-bonded compound hardly change over the wide range of $T$ considered in this work (see Supplementary Fig. 5a and b). Instead, the differences are caused by a more rapid de-population of phonons at 100 K with increasing $P$ compared to that which occurs at higher $T$, which is quantified by the Bose–Einstein distribution functions plotted in Supplementary Fig. 5c. On the other hand, the rate at which the acoustic phonon branches separate from each other as the pressure increases is roughly the same at 100 K and 1000 K. Shifting of AAO scattering rates to higher frequencies and disappearance of the AAA dip are almost purely phase space effects for BP, so they are nearly $T$-independent. Therefore, for temperatures around and above 300 K, the $T$-dependence of $\kappa(P)/\kappa_0$ is weak. However, the weakening of AAO scattering rates relative to AAA scattering rates depends strongly on $T$ below RT, resulting in a weaker $P$-dependence to $\kappa$, which eventually decreases below $\kappa_0$, as shown in Fig. 3a for $T = 100$ K. Similar behavior of $\kappa(P)$ also occurs in 3C-SiC. This unusual behavior is in contrast to that of commonly studied insulators, such as MgO, where our ab initio calculations find $\kappa(P)/\kappa_0$ increases rather than decreasing when the temperature is lowered (see Supplementary Fig. 10). The fundamental difference driving

these opposite behaviors stems from the unusually weak phase space for AAA scattering in BP and its increasingly strong competition with the AAO scattering as $T$ is lowered.

Previously, we predicted a non-monotonic pressure dependence of $\kappa$ for BAs[32], which was connected to an evolving competition between three-phonon and four-phonon scattering processes with increasing $P$. In BAs, the unusual pressure-dependent $\kappa$ occurs because of a competition between the unusually weak three-phonon scattering rates and the four-phonon scattering rates, as they evolve with pressure. Interestingly, four-phonon scattering in BP has roughly the same strength as that in BAs at ambient pressure and remains similar even to high pressures (see Supplementary Fig. 9), and both are considerably weaker than in most materials[5]. However, unlike BAs, the three-phonon scattering rates are much stronger than their four-phonon counterparts in BP. Thus, in stark contrast to BAs, the non-monotonic behavior of $\kappa(P)$ in BP arises from different physics involving competition only among three-phonon scattering channels. This competition contributes to a larger peak-$\kappa$ value in BP than in BAs (2.5x in BP vs. 1.1x in BAs at RT), and a much sharper increase in the BP $\kappa$ at low $P$ (discussed next), thus making it more accessible for experimental validation. We note that the effect of four-phonon scattering on the $\kappa(P)$ of BP is dominated by the increasing strength of the AAAA scattering channel with increasing $P$ (see Supplementary Fig. 7), but it is generally weak compared with the dominant three-phonon scattering channels, and so does not cause qualitative differences in the observed $\kappa(P)$ trends [see Figs. 1 and 3a].

**Unusually sharp rise in $\kappa(P)$ above $P = 0$.** The calculated slopes, d$\kappa$/d$P$, of the $\kappa(P)$ curves in Fig. 1 around ambient pressure are exceptionally large. Significant contributions to these large values come from the rapid increase in the intrinsic lifetimes of acoustic phonons, whose frequencies lie just below the onset of the AAO scattering. This may seem puzzling, since the AAA processes are themselves strengthening with increasing $P$. This is shown in Fig. 4a, which plots the AAA scattering rates at $P = 0$ GPa and 47 GPa for $T = 300$ K. The strengthening AAA scattering rates cause the AAA dip to become shallower, reflecting the weakening effect of the AAA selection rule caused by the increased separation of the acoustic phonon branches with increasing $P$. Such increased scattering rates should act to reduce the $\kappa$ of BP. However, the shift of the AAO processes to higher frequencies

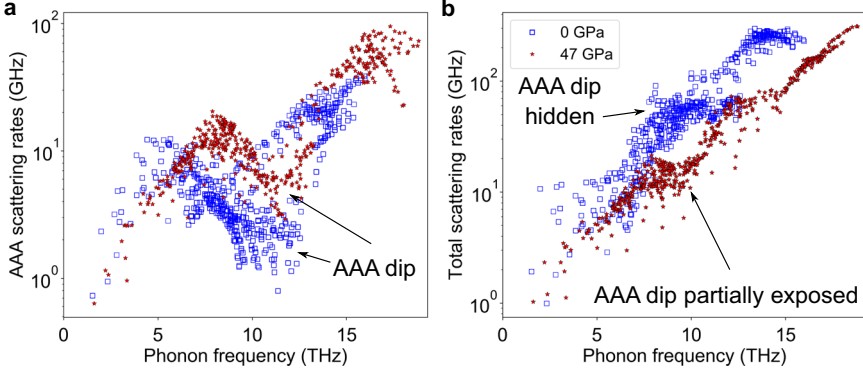

**Fig. 4 AAA and total three-phonon scattering rates for BP at different pressures. a** AAA scattering rates and **b** total three-phonon scattering rates for [11]BP at $P = 0$ GPa and $P = 47$ GPa at 300 K.

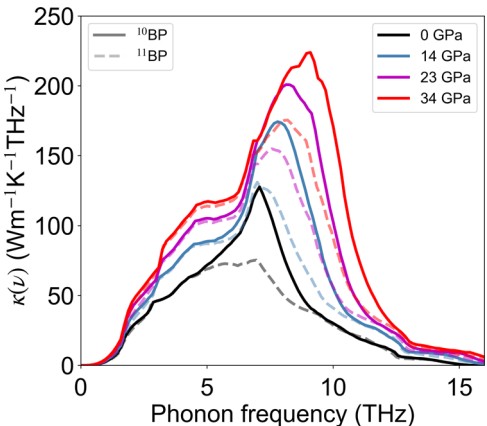

**Fig. 5 Spectral contributions to the thermal conductivity of BP.** Spectral contributions, $\kappa(\nu)$, to the thermal conductivity, $\kappa$, of [10]BP and [11]BP as a function of phonon frequency, $\nu$, at different values of $P$ at RT. The sharply peaked contributions to $\kappa$ result from the upward shift in frequency of AAO processes that increasingly expose the dip in the AAA scattering rates.

with increasing $P$ exposes portions of the AAA dip. As a result, the total scattering rates actually weaken with increasing $P$ in the 7–12 THz range, as shown in Fig. 4b. In fact, this feature (i.e., stiffer optic phonons with pressure and their effect on exposing the weak AAA scattering rates) also explains the larger calculated RT and ambient pressure $\kappa$ of [10]BP (630 Wm$^{-1}$K$^{-1}$) compared with that for [11]BP (530 Wm$^{-1}$K$^{-1}$), consistent with measured results[33,34].

The reduction in the scattering rates with increasing pressure can be seen in the spectral contributions to $\kappa$, $\kappa(\nu)$, which are plotted in Fig. 5 for [10]BP and [11]BP. The reduced total scattering rates seen in Fig. 4b are responsible for growing peaks in Fig. 5 in the ~7–12 THz frequency range, and they contribute to anomalously rapid increase of the $\kappa$ of BP with pressure. The effect is enhanced in [10]BP compared with [11]BP because the lighter [10]B mass shifts optic phonons to higher frequencies, thereby increasing the A-O gap and exposing more of the AAA dip even at 0 GPa, as seen in Supplementary Fig. 6. The low frequency enhancement of the thermal conductivity seen in Fig. 5 comes from the increase in acoustic phonon velocities and weakening three-phonon scattering rates with increasing $P$. This is the behavior typical in most materials and leads to the linearly increasing $\kappa$ with $P$.

We calculate the RT $d\kappa/dP$ values of 15.4 Wm$^{-1}$K$^{-1}$GPa$^{-1}$ ([Nat]BP), 19.1 Wm$^{-1}$K$^{-1}$GPa$^{-1}$ ([11]BP), and 24.1 Wm$^{-1}$K$^{-1}$GPa$^{-1}$ ([10]BP) at low $P$. Measurements and calculations of $d\kappa/dP$ given in

the literature are only for materials with much lower $\kappa$, such as those of interest for geophysical studies[24–27], e.g., MgO, as well as alkali halides[22,23,29]. For these materials, $d\kappa/dP$ is less than 3 Wm$^{-1}$K$^{-1}$GPa$^{-1}$, far smaller than our calculated values for BP. Therefore, to put the BP values in better context, we have calculated the RT $d\kappa/dP$ for diamond, whose $\kappa$ is the highest measured among all bulk materials, roughly five times higher than that of BP. We find RT $d\kappa/dP$ values of 21 Wm$^{-1}$K$^{-1}$GPa$^{-1}$ and 17 Wm$^{-1}$K$^{-1}$GPa$^{-1}$ for isotopically pure and naturally occurring diamond respectively. Remarkably, the calculated $d\kappa/dP$ value for [10]BP is higher than that for diamond, even though its RT $\kappa$ is much lower.

The present findings suggest that the $d\kappa/dP$ for [10]BP may be the highest value achievable for any material, a striking signature of the influence of the AAA selection rule, which is readily accessible to low pressure thermal conductivity measurements at room temperature.

## Discussion

In our prior work, we had demonstrated that the AAA selection rule triggers sharp dips in the AAA scattering phase space of other binary compounds containing B, C, and N at ambient pressure[4,5,12]. In addition, four-phonon scattering minimally influences the $\kappa$ of these materials at ambient pressures[5]. Hence, one might wonder if some of these other compounds also would have a non-monotonic pressure-dependent trend of their thermal conductivities, driven by the same mechanisms as in BP. However, unlike all of these other compounds, BP has a unique fortuitous positioning of its optic phonons relative to the acoustic phonons at ambient pressure, which sets the onset of AAO scattering exactly at the onset of the AAA dip. This feature, combined with the relative strengths of the RT AAO and AAA scattering at $P = 0$, enable the above-discussed interplay between the AAA and AAO scattering channels with changing $P$ and $T$. As described above, a sharp increase followed by a rapid decrease in the $\kappa(P)$ results. Even though the other compounds are influenced by the AAA selection rule, the relationships and interplay between AAA and AAO scattering channels are different. For example, for 3C-SiC, the frequency of onset of the AAO scattering channel lies below the AAA dip at ambient pressure, and the RT AAO scattering rates remain stronger than the AAA scattering rates over the AAA dip until high pressure, by which time the dip has nearly disappeared. As a result, 3C-SiC shows a mainly increasing $\kappa(P)$ with increasing $P$. Results for 3C-SiC are given in the Supplementary Figs. 11–13. Another example is cubic boron nitride (BN). For BN, the A-O gap vanishes because of the small N to B mass ratio (1.3), so even application of high pressure does not open an A-O gap to expose the AAA dip

(see Supplementary Figs. 15a and 16). Furthermore, four-phonon scattering rates are also weaker than three-phonon scattering rates in BN at 300 K and over a range of pressures (see Supplementary Fig. 17). As a result, BN shows the typical monotonically increasing thermal conductivity with $P$, as shown in our prior publication[32] and also reproduced as Supplementary Fig. 14. Results for BN are given in Supplementary Figs. 14–17. Thus, BP appears to be the ideal candidate material to interrogate the impact of the AAA selection rule on phonon–phonon scattering through pressure and temperature tuning.

We note that high quality samples of $^{Nat}$BP and isotopically enriched $^{11}$BP and $^{10}$BP can now be fabricated[33,34]. The intrinsic carrier densities at room temperature and below are low because of the large energy gap (~2 eV). Carrier concentrations measured in some of these samples are less than $10^{18}$ cm$^{-3}$. Thus, we expect that electron–phonon scattering has minimal influence on the $\kappa$ of BP in such samples. In fact, our first principles calculations have shown quantitative agreement with the measured thermal conductivities of $^{Nat}$BP and isotopically enriched $^{11}$BP and $^{10}$BP without including electron–phonon scattering[33,34].

Finally, we note that BP remains stable in the zinc blende structure to at least 110 GPa[38]. Thus, all of the anomalous features in the pressure-dependent $\kappa$ of BP identified above occur within the region of stability in the zinc blende structure.

In summary, through ab initio calculations we have shown that the hidden influence of a weak three-phonon scattering channel arising from a selection rule on anharmonic decay of acoustic phonons can be revealed through examination of the pressure and temperature dependence of the thermal conductivity. An excellent candidate material for this investigation is boron phosphide (BP), which displays anomalous features including a rapid rise and a peak in $\kappa$ with increasing pressure. These anomalous behaviors in BP occur at pressures and temperatures that are readily accessible to measurements. The present work gives fundamental insights into the impact of selection rules on anharmonic phonon scattering and reveals an anomalous behavior of phonon thermal transport in solids under pressure.

## Methods

We solve the linearized Boltzmann transport equation for phonons for the non-equilibrium distribution function, $n_\lambda = n_\lambda^0 + n_\lambda^1$, resulting from a small applied temperature gradient, $\nabla T$, where $n_\lambda^0$ is the Bose distribution function, and $n_\lambda^1$ is the small deviation from $n_\lambda^0$ generated by $\nabla T$. Here, $\lambda \sim (\mathbf{q}, j)$ designates the phonon mode, where $j$ is the phonon branch and $\mathbf{q}$ is the phonon wave vector. The linearized Boltzmann equation is:

$$\mathbf{v}_\lambda \cdot \nabla T \frac{\partial n_\lambda^0}{\partial T} = \frac{\partial n_\lambda}{\partial t}\bigg|_{\text{collisions}} \tag{1}$$

where $\mathbf{v}_\lambda$ is the group velocity of the phonon mode $\lambda$, and the right-hand side is the collision term, which includes three-phonon scattering, four-phonon scattering and phonon scattering by the isotopic mass disorder. Harmonic interatomic force constants (IFCs), and anharmonic third- and fourth-order IFCs are determined using the Density Functional Theory (DFT) as implemented in Quantum Espresso[39,40]. Expressing $n_\lambda^1 = n_\lambda^0(n_\lambda^0 + 1)\mathbf{F}_\lambda \cdot (-\nabla T)$ allows Eq. (1) to be recast into an integral equation for the function, $\mathbf{F}_\lambda$, which is solved numerically. The thermal conductivity is then obtained as:

$$\kappa_{\alpha\beta} = \frac{k_B T^2}{V} \sum_\lambda \frac{\partial n_\lambda^0}{\partial T} v_{\lambda,\alpha} F_{\lambda,\alpha} \tag{2}$$

where $k_B$ is the Boltzmann constant, $V$ is the crystal volume, and $\kappa_{\alpha\beta}$ is the thermal conductivity tensor for heat current flow along the Cartesian direction, $\alpha$, resulting from a temperature gradient along the direction, $\beta$. For the cubic structures considered in the present work, the thermal conductivity tensor is diagonal: $\kappa_{\alpha\beta} = \kappa \delta_{\alpha\beta}$.

Further details of the computational approach including computation scheme for IFCs, detailed expressions for the phonon scattering rates and implementation of the solution of the phonon Boltzmann equation are given in ref. [31]. In particular, the three-phonon scattering probabilities, given by:

$$\begin{aligned}
W_{\lambda\lambda_1\lambda_2}^{(+)} &= \frac{2\pi}{\hbar^2}\left|\Xi_{\lambda\lambda_1(-\lambda_2)}\right|^2 \left(n_{\lambda_1}^0 - n_{\lambda_2}^0\right)\delta\left(\omega_\lambda + \omega_{\lambda_1} - \omega_{\lambda_2}\right) \\
W_{\lambda\lambda_1\lambda_2}^{(-)} &= \frac{2\pi}{\hbar^2}\left|\Xi_{\lambda(-\lambda_1)(-\lambda_2)}\right|^2 \left(1 + n_{\lambda_1}^0 + n_{\lambda_2}^0\right)\delta\left(\omega_\lambda - \omega_{\lambda_1} - \omega_{\lambda_2}\right)
\end{aligned} \tag{3}$$

and the four-phonon scattering probabilities, given by:

$$\begin{aligned}
Y_{\lambda\lambda_1\lambda_2\lambda_3}^{(1)} &= \frac{2\pi}{\hbar^2}\left|\Xi_{\lambda(-\lambda_1)(-\lambda_2)(-\lambda_3)}\right|^2 \frac{n_{\lambda_1}^0 n_{\lambda_2}^0 n_{\lambda_3}^0}{n_\lambda^0}\delta\left(\omega_\lambda - \omega_{\lambda_1} - \omega_{\lambda_2} - \omega_{\lambda_3}\right) \\
Y_{\lambda\lambda_1\lambda_2\lambda_3}^{(2)} &= \frac{2\pi}{\hbar^2}\left|\Xi_{\lambda\lambda_1(-\lambda_2)(-\lambda_3)}\right|^2 \frac{\left(1+n_{\lambda_1}^0\right)n_{\lambda_2}^0 n_{\lambda_3}^0}{n_\lambda^0}\delta\left(\omega_\lambda + \omega_{\lambda_1} - \omega_{\lambda_2} - \omega_{\lambda_3}\right) \\
Y_{\lambda\lambda_1\lambda_2\lambda_3}^{(3)} &= \frac{2\pi}{\hbar^2}\left|\Xi_{\lambda\lambda_1\lambda_2(-\lambda_3)}\right|^2 \frac{\left(1+n_{\lambda_1}^0\right)\left(1+n_{\lambda_2}^0\right)n_{\lambda_3}^0}{n_\lambda^0}\delta\left(\omega_\lambda + \omega_{\lambda_1} + \omega_{\lambda_2} - \omega_{\lambda_3}\right)
\end{aligned} \tag{4}$$

which directly enter the solution of the Boltzmann equation, contain the energy and quasi-momentum conservation constraints for phonon scattering, apart from the temperature dependence through the Bose factors. In these equations, $\Xi_{\lambda\lambda_1\lambda_2}$ and $\Xi_{\lambda\lambda_1\lambda_2\lambda_3}$ are the three-phonon and four-phonon matrix elements respectively, and $\delta(\cdot)$ is the Dirac delta function. All the calculations in this study are performed using norm-conserving pseudopotentials with the local density approximation (LDA) for the exchange correlation. The converged parameters used in the first principles calculations in this work for BP, 3C- SiC and BN, such as the energy cutoffs for the DFT calculations, $\mathbf{q}$-grids used in the solution of the Boltzmann equation and the cutoffs for the harmonic, cubic and quartic IFCs, can be found in refs. [5,32].

The pressure in our calculations is calculated by obtaining the derivative of the Helmholtz free energy, correct to fourth-order in anharmonicity ($F_{\text{4th-order}}$), with respect to crystal volume ($V$) at each temperature ($T$) using the expression:

$$P(a) = -\frac{\partial F_{\text{4th-order}}}{\partial V}\bigg|_{T,a} \approx -\frac{F_{\text{4th-order}}(a+\Delta a) - F_{\text{4th-order}}(a-\Delta a)}{V(a+\Delta a) - V(a-\Delta a)}\bigg|_T \tag{5}$$

where $a$ is the crystal lattice constant and $\Delta a \sim 0.05\%$ of $a$. The complete expression for the Helmholtz free energy, correct to fourth-order in anharmonicity of the crystal potential, can be found in section 10 of the Supplemental Materials in ref. [32]. The pressure vs. volume relationship calculated for BP is presented in Supplementary Fig. 2a. From it, we obtain a bulk modulus of 170.5 GPa, which is within 1% of the measured value in the literature[41].

## Data availability
The data supporting the findings of this work are available from the corresponding author upon reasonable request.

## Code availability
All formulations and algorithms necessary to reproduce the results of this study are described in the Methods section and in ref. [31].

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

## Acknowledgements

This work was supported in part by the Office of Naval Research, USA, under a Multidisciplinary University Research Initiative, grant No. N00014-16-1-2436. N. K. R. acknowledges support by a start-up fund from the Indian Institute of Science, Bangalore, India. We thank Prof. Bing Lv of the University of Texas at Dallas for providing estimates of the measured carrier concentrations in the BP samples grown in his laboratory. We also acknowledge computational support from the Boston College Linux Cluster.

## Author contributions

N.K.R and D.B. originated the research. N.K.R. performed the ab initio calculations. N.K.R. and D.B. analyzed the results and wrote the manuscript. Both authors studied, commented on, and edited the manuscript.

## Competing interests

The authors declare no competing interests.
