## [Peer Review File · Nature Communications]

REVIEWER COMMENTS

Reviewer #1 (Remarks to the Author):

The authors report reports the non-monotonic behaviour of thermal conductivity (κ) with temperature and pressure in boron phosphide (BP) using first principle calculations. The claim to observe sharpest increase in κ with increasing pressure. The author have observed similar non-linear increase in BAs, however, they claim the origin to be quite different. This is very puzzling as it brings in an ad-hocness in the process of calculation of thermal conductivities in this exciting class of materials. Further, authors insights are merely numerical observations, and have no origin in the physics of the behaviour. Therefore, manuscript leaves behind several unclear doubts, which in my opinion can only be resolved by experiments. Therefore, I would recommend the publication of this manuscript in Nature Communications.

Below are some other comments:

1. What is the reason that four-phonon scattering to be relatively weaker in BP than in boron arsenide (BAs) Ref. 31?
2. Authors report by isoelectronic substitution of arsenic by phosphorus a drastic change in strength of three-phonon and four-phonon process. Why these changes are so drastic? How this interchange affecting the scattering processes?
3. It is also expected that BP will have sharper rise and higher peak for thermal conductivity than that was in BAs, since BP is lighter in mass than BAs.
4. The claim that BP may have the steepest peak, did the authors check the behaviour of boron nitride (BN)? Is there a trend in the strength of four and three phonon scattering?
5. Is there any physics for including only three-phonon processes or both three-phonon and four-phonon process in any given material. How will we decide that for a given material?
6. Is there any physics behind so called selection rule reported by the authors.

Reviewer #2 (Remarks to the Author):

This manuscript presents an investigation on the thermal transport properties of boron phosphide based on the Boltzmann transport equation theory combined with first-principle calculations. The obtained results show that the thermal conductivity of boron phosphide increases first, and then decreases with increasing pressure, which violates the traditional knowledge that pressure normally enhances thermal conductivity. The explanation for the unusual behavior is believed to be the evolving interplay between AAA and AAO scattering channels with varying pressure. Moreover, it also found that the effect of four-phonon scattering on thermal conductivity is relatively weak at all pressures in boron phosphide. Generally speaking, this paper made a deep investigation towards that the role of selection rules plays in thermal transport. It can be published in this journal. The logical illustration is acceptable. However, there are several problems still needed a clear explanation.

1. The atomic configuration of boron phosphide is necessary to be provided, which helps the readers intuitively understand the structure-property relation. How the crystal volumes/structure that directly affects the specific heat in Equ.(2) change with increasing pressure?
2. A-O gap is important for competitive relations between 3- and 4- phonon scattering process. I suggest the authors provide a detailed data of A-O gap under the different pressures to clear the relations between A-O gap and different phonon scattering process.
3. Fig. 2 show that the scattering rates of AAA process at 46 GPa have almost one magnitude enhancement than that of 0 GPa. However, Fig. 4 shows that the phonons in range from 0~15 THz contributed thermal conductivity are enhanced under the pressure from 0~34 GPa. In general, the

increasing scattering rates will decrease the thermal conductivity. So, what reason could be response for the increasing of thermal conductivity?

4. To better understand the results in Fig. 3(a), the effect of temperature on the phonon dispersion should be discussed in detail.

5. Could the authors evaluate how the electron-phonon interaction affects the obtained thermal conductivity in boron phosphide?

6. The authors claimed that the influence of the selection rule on thermal conductivity was long thought to be negligible since intrinsic thermal resistance is dominated by scattering processes involving phonons in different branches, particularly at higher temperatures. Actually, the importance of selection rule has been involved in many previous studies (such as *J. Phys.: Condens. Matter* 2020, 32, 153002). It is suggested to add a brief discussion on the effect of the breaking of the selection rule on phonon transport and the related reference.

7. As mentioned in article [*Phys. Rev. B*, 78 (22), 224303, 2008] and [*Adv. Funct. Mater.*, 30 (8), 1903873, 2020], the lattice thermal conductivity with consideration of full anharmonic effect can also be calculated by atomic Green's function method. What different between AGF method and the theoretical method presented in this work? For examples, whether the contribution of high order Feynman diagram of three- and four- phonon interactions can also be considered in this method?

Reviewer #3 (Remarks to the Author):

The authors studied the pressure and temperature dependence of thermal conductivity for BP and found anomalous features including an exceptionally fast rise and a peak with increasing pressure. They stated that the invisible impact of selection rules on three-phonon scattering can be exposed through this kind of anomalous signatures.

This manuscript is well organized, clearly written and scientifically sound. The interesting results occur at experimentally accessible conditions and may motivate experimental efforts to observe the predicted novel behavior. I recommend it being published in *Nature communications* after the following revisions.

1. The highest pressure and temperature that the authors studied is over 100 GPa and up to 1000 K. Does BP keep the the same structure at such high pressure and temperature? The related discussion should be added.

2. It is very interesting to see that the thermal conductivity has a peak as the pressure increases. But the same trend was also predicted for BeSe (PRB 91, 121202(R) 2015) by the same authors, is the reason discussed here in BP suitable for BeSe?

3. In page 4, 10B in the sentence of (3) B atoms isotopically enriched to 100 % 10B (10BP) should be changed to 10B.

Rebuttal/Response report for the manuscript titled, "Exposing the hidden influence of selection rules on phonon-phonon scattering by pressure and temperature tuning", by Navaneetha K. Ravichandran and David Broido.

We thank all three referees for taking the time to carefully read our manuscript and provide valuable suggestions and comments. All three referees have recommended publication of this manuscript in Nature Communications after we address their minor concerns and incorporate their suggestions. We have now addressed all of the referees' concerns and suggestions in the revised manuscript, and also provided a point-wise response to the referees' comments below. Our responses are in blue. Changes/additions to the text of the revised manuscript are in green.

Reviewer #1 (Remarks to the Author):

The referee says, "The authors report reports the non-monotonic behaviour of thermal conductivity (κ) with temperature and pressure in boron phosphide (BP) using first principle calculations. The claim to observe sharpest increase in k with increasing pressure. The author have observed similar non-linear increase in BAs, however, they claim the origin to be quite different. This is very puzzling as it brings in an ad-hocness in the process of calculation of thermal conductivities in this exciting class of materials. Further, authors insights are merely numerical observations, and have no origin in the physics of the behaviour. Therefore, manuscript leaves behind several unclear doubts, which in my opinion can only be resolved by experiments. Therefore, I would recommend the publication of this manuscript in Nature Communications."

Response: We thank the referee for recommending the publication of our manuscript in Nature Communications. While we agree with the referee that these rigorous first-principles predictions need and will motivate future experimental confirmation (beyond the scope of the current study), we respectfully disagree with a couple of points that the referee brings up in her/his summary remark.

First, we do not think there is any ad-hocness in the thermal conductivity calculations. We have proved through our prior publications (Refs. [5, 19, 31-33]) that the unified first principles framework (developed by us in Ref. [31]) that is utilized in the current study, is fully predictive and quantitatively reproduces the measured thermal conductivities and thermodynamic properties as functions of temperature and pressure for a broad class of materials, *without any ad-hoc/adjustable parameters*. Moreover, we have used the same unified first principles framework (including phonon renormalization, Helmholtz free energy calculations, pressure calculations and solutions of the Boltzmann equation for phonon transport including three-phonon, four-phonon and phonon-isotope scattering) for both the cases of BAs (in Ref. [32]) and BP (in the current manuscript). Therefore, the calculations are never adjusted in any ad-hoc manner.

Second, the fact that we observe drastic quantitative differences in $k(P, T)$ for BAs and BP, and the fact that we are able to clearly link these differences to different competition mechanisms among different phonon-phonon scattering channels (i.e., 3-ph. vs. 4-ph. in BAs, compared to AAA (3-ph.) vs AAO (3-ph.) in BP) using the same numerical framework, clearly shows that the observed differences originate in the different physics of phonon scattering mechanisms in these two materials (BAs vs. BP). Thus, the predicted anomalous pressure-dependencies of the thermal conductivity of BP are firmly originating from the physics of phonon-phonon scattering mechanisms, and are not just numerical observations.

We now address the referees' concerns in the point-wise responses below:

1. The referee asks, "What is the reason that four-phonon scattering to be relatively weaker in BP than in boron arsenide (BAs) Ref. 31?"

Response: Four-phonon scattering in BAs is, in fact, comparable in strength to that in BP, and both are quite weak, as we have discussed in Refs. [5, 33]. This is shown in Fig. R1 (red points). The reason that the *effect* of four-phonon scattering on the thermal conductivity (k) of BAs is much stronger than in BP (55% vs. 5% reduction in k with the inclusion of four-phonon scattering at $T = 300$ K and $P = 0$ GPa) is that BAs has anomalously weak three-phonon scattering strength in the 4-8 THz frequency region (labeled AAA dip in Fig. R1). On the other hand, Fig. R1 shows that in BP, three-phonon scattering is much stronger than four-phonon scattering, so the effect of four-phonon scattering on the k of BP is minimal.

Figure R1: Comparison of three-phonon and four-phonon scattering rates for BP (a) and BAs (b) at 300 K and 0 GPa. The sharp dip in the three-phonon scattering rates due to the AAA selection rule is seen in BAs but hidden in BP at ambient pressure by strong AAO scattering.

A natural question that arises at this point is, "Why are the total three-phonon scattering rates anomalously weak in BAs, but are much stronger in BP?" As explained in the manuscript, three-phonon scattering between two acoustic (A) phonons and one optic (O) phonon (AAO scattering) occurs in BP but is almost absent in BAs. For a wide frequency range, the AAO scattering rates are much larger than the scattering between three acoustic phonons (AAA scattering) in BP. The AAA scattering rates in both BP and BAs show sharp dips because of the action of the AAA selection rule discussed in our paper. The sharp dip is clearly seen in the total three-phonon scattering rates in BAs (see Fig. R1), with three-phonon scattering rates dipping below 1 GHz around a phonon frequency of ~ 8 THz, even smaller than the four-phonon scattering rates in that frequency range. Phonons in this frequency range have large lifetimes and give correspondingly large contributions to the thermal conductivity of BAs. In BP, the sharp dip that occurs in the AAA scattering rates is *hidden* by the much stronger AAO scattering (see Fig. 2 (b) in the revised main manuscript; compare red (AAA) and green (AAO) points in the 0 GPa plot). A central finding in our manuscript is that this otherwise hidden effect of the AAA selection rule on thermal transport in BP can be uncovered through the application of pressure, and that it manifests itself through the anomalous pressure dependent thermal conductivity that we have identified in BP.

Figure R2 shows the four-phonon scattering rates of BAs and BP at different temperatures and pressures: (300 K, 0 GPa), (300 K, ~ 33 GPa), (300 K, ~ 75 GPa), (750 K, 0 GPa), (750 K, ~ 33 GPa) and (750 K, ~ 75 GPa); it shows that the four-phonon scattering rates of BAs and BP under each of these six conditions remain similar in magnitude.

Figure R2: Comparison of four-phonon scattering rates for BP and BAs at different temperatures and pressures. Overall, the four-phonon scattering rates are comparable in magnitude under all of the conditions considered in this work for BP and BAs.

To address the referee's comment and clarify this point, we have added Figs. R1 and R2 to the Supplementary Materials (new Figs. S9 and S10) and we have included the following text on pg. 9 of the revised manuscript:

"Previously, we predicted a non-monotonic pressure dependence of k for BAs [32], which was connected to an evolving competition between three-phonon and four-phonon scattering processes with increasing P . In BAs, the unusual pressure dependent k occurs because of a competition between the unusually weak three-phonon scattering rates and the four-phonon scattering rates as they evolve with pressure. Interestingly, four-phonon scattering in BP is roughly the same strength as that in BAs (see Fig. S10 in the Supplementary Materials section S8), and both are considerably weaker than in most materials [5]. However, unlike BAs, the three phonon scattering rates are much stronger than their four-phonon counterparts in BP. Thus, in stark contrast to BAs, the non-monotonic behavior of $k(P)$ in BP arises from different physics..."

2. The referee asks, "Authors report by isoelectronic substitution of arsenic by phosphorus a drastic change in strength of three-phonon and four-phonon process. Why these changes are so drastic? How this interchange affecting the scattering processes?"

Response: As described above, the strength of four-phonon scattering processes in BP and BAs are similar in magnitude.

On the other hand, as the referee correctly points out, the strength of three-phonon scattering rates are drastically different between BP and BAs at ambient conditions. This sharp difference comes from the differences in the frequency gaps between acoustic and optic phonons (A-O gap) in BP and BAs, as described in our earlier study (Ref. [5]). A selection rule on scattering between two acoustic phonons and one optic phonon (AAO selection rule) requires that the frequencies of participating acoustic phonons must be at least as large as the frequency of the A-O gap. BAs has a large A-O gap due to the large As to B mass ratio (~ 7). Then, the AAO selection rule almost completely freezes out any AAO scattering channels in BAs. On the other hand, the A-O gap is not large enough in BP under ambient conditions due to the smaller P to B mass ratio (~ 2.6). Therefore, AAO scattering occurs in BP, and AAO scattering rates are strong beyond ~ 5 -6 THz (which is the A-O frequency gap under ambient conditions). Thus, the differences in the strength of three-phonon scattering rates by the iso-electronic substitution of arsenic by phosphorus (going from BAs to BP) come from the strong AAO scattering rates in BP, driven by the differences in the phonon dispersions. In fact, upon turning off the AAO scattering processes artificially (as a numerical experiment), our calculations show that (i) the three-phonon limited k of BP exceeds $6000 \text{ Wm}^{-1}\text{K}^{-1}$, since the AAA three-phonon scattering channels, that have been weakened by the AAA selection rule, are now fully exposed, similar to BAs, and (ii) inclusion of four-phonon scattering dramatically reduces the k of BP (to around $2600 \text{ Wm}^{-1}\text{K}^{-1}$), again similar to BAs. So, in brief, the isoelectronic substitution As by P turns on strong AAO scattering that hides the striking similarities between BP and BAs.

To emphasize this important point, we have revised the discussion of turning off AAO processes on pg. 6 of the revised manuscript to read:

Thus, the isoelectronic substitution of As with P activates strong AAO scattering that hides the otherwise similar behavior BP has to BAs. A comparison of room temperature three-phonon and four-phonon scattering rates in BP and BAs is given in the Supplementary Materials section S7 (Fig. S9). To illustrate the importance of the AAO processes in BP, we have performed calculations in which they have been artificially turned off. These calculations show that (i) the three-phonon limited k of BP then increases by over an order of magnitude, exceeding $6000 \text{ Wm}^{-1}\text{K}^{-1}$, and (ii) inclusion of four-phonon scattering dramatically reduces the k of BP to around $2600 \text{ Wm}^{-1}\text{K}^{-1}$. The ultrahigh thermal conductivity and its sharp reduction from four-phonon scattering are qualitatively similar to the behavior in BAs.

3. The referee states, "*It is also expected that BP will have sharper rise and higher peak for thermal conductivity than that was in BAs, since BP is lighter in mass than BAs.*"

Response: We respectfully disagree with this point by the referee. It is true that, among the group IV and group III-V materials, the lighter compounds often have higher phonon frequencies and group velocities. However, by the referee's reasoning, then boron nitride (BN) and diamond should have more rapid rises in $k(P)$ than does BP since they are lighter. In fact, according to our calculations, they do not, even though their thermal conductivities are several times larger than that of BP. Figure R3 shows this comparison for BP and BN. The sharpness of the $k(P)$ rise in BP that we find is extremely unusual. It derives from a unique combination of the material properties and the interplay of AAA and AAO scattering rates as they evolve with pressure. Furthermore, in most materials, the thermal conductivity increases monotonically with pressure; there is no peak. BAs and BP are anomalous in showing the peak behavior. As we have detailed in the manuscript and in the response below, the peak behaviors of $k(P)$ in BAs and BP arise from different physics; so the higher peak in BP compared with BAs cannot be connected to BP being lighter in mass.

We have added Fig. R3 to the Supplementary Materials (as Fig. S15) to emphasize this point. The figure shows that the $k(P)$ of BP has a much faster increase with P compared with BN. [The $k(P)$ curves at 300 K for BN has been calculated and reported by us in our earlier work (Ref. [32]).]

Figure R3: Three-phonon (a) and 3+4-phonon (b) limited k of BN and BP as functions of pressure at room temperature, relative to their corresponding values under ambient pressure (k_0). Even though BN is lighter in mass, BP has the largest rate of rise in k .

4. The referee asks, "*The claim that BP may have the steepest peak, did the authors check the behaviour of boron nitride (BN)? Is there a trend in the strength of four and three phonon scattering?*"

Response: As mentioned in the previous point, we have indeed performed calculations of three-phonon and 3+4-phonon limited k for BN at 300 K as a function of pressure, and have reported these calculations in our prior work (Ref. [32], Fig. 1c).

Briefly, as shown in Fig. R3, the rise in k of BN with pressure at 300 K and at low pressure is not as steep as that in BP (with or without four-phonon scattering included) even though BN has over three times higher k than BP at ambient pressure and temperature. Also, the k of BN shows no peak; instead it increases monotonically with pressure.

Even up to high pressures, there are no frequency gaps created between the acoustic phonons and the optic phonons in BN. Hence, the AAO scattering rates remain dominant over most of the frequency spectrum, especially over the weakest AAA scattering rates even up to high pressures. An example at 22 GPa is shown in Fig. R4. The observed weaker rise in k in BN is predominantly caused by a slight stiffening of the acoustic phonons (resulting in higher group velocities of the heat-carrying acoustic modes), along with a slight weakening of the overall three-phonon scattering rates (driven predominantly by the reduced occupation numbers of the stiffening optic phonons with pressure). Hence, the pressure dependence of k for BN occurs for the same reasons as in other conventional materials like MgO. In particular, while the AAA scattering rates in BN are weakened by the AAA selection rule as in BP and BAs, this effect is never exposed in the $k(P)$ of BN because the A-O gap is zero in the phonon dispersions, so the AAO scattering rates dominate even at high pressures. The behavior is shown in Fig. R5.

In stark contrast, the stiffening of the optic phonons in BP by the application of pressure shifts the A-O gap to higher frequencies. Correspondingly, the frequency of onset for the AAO processes increases, according to the AAO selection rule, thereby exposing the influence of the AAA selection rule (i.e. weak AAA scattering rates), which is hidden at zero pressure, as described in the main manuscript.

Figure R4: (a) Phonon dispersions and (b) three-phonon scattering rates of BN at 300 K and at two different pressures - 0 GPa and 22 GPa. There is no A-O frequency gap in BN, even at high pressures. The k of BN increases with increasing pressures (at low pressure) due to the slightly stiffening acoustic phonon frequencies (which results in a slight increase in the acoustic phonon group velocities) and a slightly weakening three-phonon scattering rates with increasing pressure.

By comparison, three-phonon scattering strengths in BN are similar to those in BP; both are far stronger than those in BAs. Furthermore, the four-phonon scattering rates are significantly weaker relative to the three-phonon scattering rates, as shown in Fig. R6, and they are similar in magnitude to those of BP and BAs. Hence, the effect of four-phonon scattering on the $k(P)$ curves of BN are very minimal.

We have now included these plots (Figs. R4-R6) in the Supplementary Material (as Figs. S16-S18) highlighting these important distinctions (and, in some cases, similarities) between the pressure-dependent k and the scattering rates of BP and BN. We have also added the following text to the discussion section of the main manuscript after the discussion of 3C-SiC on pg. 13:

Results for 3C-SiC are given in Supplementary Materials section S10. Another example is cubic boron nitride (BN). For BN, the A-O gap vanishes because of the small N to B mass ratio (1.3) so even application of very high pressure does not open an A-O gap to expose the AAA dip (see Figs. S16 (a) and S17 in the Supplementary Materials section S11). Furthermore, four phonon scattering rates are also significantly weaker than three-phonon scattering rates in BN at 300 K and over a range of pressures (see Fig. S18 in the Supplementary Materials section S11). As a result, BN shows the typical monotonically increasing thermal conductivity with P , as shown in our prior publication [32] and also reproduced as Fig. S15 in the Supplementary Materials section S11.

5. The referee asks, "Is there any physics for including only three-phonon processes or both three-phonon and four-phonon process in any given material. How will we decide that for a given material?"

Response: In general, three-phonon scattering dominates the behavior:

- (1) At low to moderate temperatures (compared to e.g., the Debye temperature of the material)
- (2) For strongly bonded materials.

At high temperatures or for weakly bonded compounds (e.g. the alkali halides such as NaCl), the vibrating atoms have large amplitude displacements and so sample larger regions of anharmonic bonding. In such cases, the four-phonon scattering can play an important role in limiting the phonon transport.

Figure R5: AAA, AAO and AOO three-phonon scattering rates for cubic BN as functions of phonon frequencies at different pressures and room temperature. Similar to BP, the AAA scattering rates are weakened by the AAA selection rule due to the bunching-together of the acoustic phonons, and are anomalously weak (highlighted by the black ovals), as described in the manuscript. However, unlike BP, the AAO scattering rates completely dominate these anomalously weak AAA scattering rates, since there is no frequency gap between the acoustic and the optic phonons (A-O gap) in BN. This results in overall strong three-phonon scattering rates and negligible competition between AAA and AAO scattering channels with increasing pressure.

We note that inclusion of four-phonon scattering improves quantitative agreement with experiment even in strongly bonded compounds [5, 19, 31-33]. For some compounds with large mass ratio of the constituent atoms, such as BAs, AlSb and InP, these general guidelines fail catastrophically, not because four-phonon scattering is strong, but because three-phonon scattering becomes unusually weak [5].

The temperature-dependence of four-phonon scattering rates is much stronger than that of the three-phonon scattering rates, as we have shown in our prior work (e.g., Supplementary Figure 12 in Ref. [32]). Therefore, for those materials with comparable three-phonon and four-phonon scattering rates at room temperature (e.g., NaCl [31]), the four-phonon scattering rates become significantly weaker than their three-phonon counterparts at low temperatures.

Figure R6: Three-phonon vs. four-phonon scattering rates for cubic BN as functions of phonon frequencies at different pressures and room temperature. Four-phonon scattering rates are much weaker than the three-phonon scattering rates at room temperature and all pressures considered here.

6. The referee asks, "*Is there any physics behind so called selection rule reported by the authors.*"

Response: The key physics stems from the requirement that anharmonic phonon decay processes must conserve energy and quasi-momentum. The selection rules follow directly from these conservation conditions, and they constrain certain anharmonic decay channels. The two selection rules on which we focus in the present work are what we call the AAA selection rule and the AAO selection rule.

The AAA selection rule states that an acoustic phonon cannot decay anharmonically into two others with higher phase velocity (see Ref. [11]); it prevents almost all anharmonic decay of phonons within the same dispersive acoustic phonon branch (see Refs. [3-5, 7-13]). Thus, AAA processes can occur only if they involve phonons in different acoustic branches. In some crystals including

BAs and BP, the chemical bonding causes acoustic phonon branches to come close together in certain high frequency regions of the Brillouin zone. For phonons in such regions, their opportunities for anharmonic decay via AAA processes become severely limited since decay into other phonon branches is approaching decay within a single branch, which is forbidden by the AAA selection rule.

The AAO selection rule states that acoustic phonons cannot participate in AAO processes if their frequencies, ω , are smaller than that of the A-O gap, ω_{A-O} . Thus, this selection rule dictates the frequency onset of AAO processes: $\omega \geq \omega_{A-O}$

This physics is conveyed in the theory through the phonon-phonon scattering probabilities. For example, the expressions for the three-phonon scattering probabilities are given by:

$$W_{\lambda\lambda_1\lambda_2}^{(+)} = \frac{2\pi}{\hbar^2} |\Xi_{\lambda\lambda_1(-\lambda_2)}|^2 (n_{\lambda_1}^0 - n_{\lambda_2}^0) \delta(\omega_\lambda + \omega_{\lambda_1} - \omega_{\lambda_2})$$

$$W_{\lambda\lambda_1\lambda_2}^{(-)} = \frac{2\pi}{\hbar^2} |\Xi_{\lambda(-\lambda_1)(-\lambda_2)}|^2 (1 + n_{\lambda_1}^0 + n_{\lambda_2}^0) \delta(\omega_\lambda - \omega_{\lambda_1} - \omega_{\lambda_2})$$

where, $W^{(+)}$ and $W^{(-)}$ correspond to two different *types* (in-scattering and out-scattering) of three-phonon processes. In the equations, $\lambda \sim (\mathbf{q}, j)$ designates a phonon mode (wavevector, \mathbf{q} and branch index, j), n_λ^0 is the Bose factor of the mode λ , and the $|\Xi|^2$ are three-phonon matrix elements. Selection rules reflecting an underlying symmetry in the system can cause the matrix elements to vanish. However, the selection rules discussed in our work are different. They appear through the energy (or frequency) conservation constraint explicitly in the form of a Dirac delta function, (the quasi-momentum conservation has already been implicitly included).

For some binary compounds with large heavy-to-light atom mass ratios, phonon dispersions have an A-O gap that is larger than the highest acoustic phonon frequency. Then, according to the AAO selection rule, no AAO scattering processes are possible. This is essentially what happens in BAs. The smaller P to B mass ratio in BP results in an A-O gap that is smaller than the highest acoustic phonon frequency. So, the acoustic phonons with frequencies larger than the A-O gap can participate in the AAO scattering channel, while those with frequencies smaller than the A-O gap are completely forbidden from participating in the AAO scattering channel, according to the AAO selection rule.

To clarify the effect of the selection rules on the AAA and AAO scattering rates, we have added Fig. R7 as Fig. S4 to be part of the Supplementary Materials section S2. The figures illustrate schematically, the restrictions imposed on the AAA and AAO processes when selection rules are activated by features in the phonon dispersions. Panel (a) shows that only *interbranch* AAA processes are generally allowed by the AAA selection rule. Panel (b) shows that acoustic phonon decay is inhibited by the AAA selection rule for high frequency phonons in regions where phonon branches come close together. This behavior is responsible for the AAA dips in BP and BAs. Panel (c) shows that AAO processes cannot occur for sufficiently large A-O frequency gap. This is the case for BAs. Panel (d) shows that AAO processes *can* occur for materials having a smaller A-O gap. This is the case for BP.

As we have described in detail the physics of the selection rules on phonon scattering processes in our prior (recent) publication (Ref. [5]), to avoid repeating this published work, we have not made any significant changes in this regard to our current manuscript. Instead, we have cited Ref. [5] at appropriate places in the revised manuscript to ensure that the reader is fully aware of our prior work and conclusions therein.

Figure R7: Schematic of the effects of AAA and AAO selection rules for hypothetical phonon dispersions. (a) Three-phonon anharmonic decay within a single acoustic (A) phonon branch ($T_1 + T_1 \leftrightarrow T_1$) is forbidden by the AAA selection rule (marked with X). The interbranch process ($T_1 + T_2 \leftrightarrow L$) is allowed. (b) Hypothetical dispersions where L and T_2 branches approach each other and are degenerate at the point 2; a phonon at this point cannot decay within either L or T_2 branch, and the other AAA decay channels such as $T_1 + T_2/L \leftrightarrow L$ are weak. (c) If the frequency gap between the acoustic and the optic (O) phonons (A-O gap) is large, anharmonic decay involving two acoustic and one optic phonons (AAO process) is forbidden. Phonons around point 2 will have large intrinsic lifetimes, since the AAO processes are forbidden and the AAA processes are weak. (d) If the A-O gap is small, then the AAO processes can be strong, such as the $T_1 + T_2/L \leftrightarrow O$ process at points 1, 2 and 3 shown in the figure.

Reviewer #2 (Remarks to the Author):

The referee says, "This manuscript presents an investigation on the thermal transport properties of boron phosphide based on the Boltzmann transport equation theory combined with first-principle calculations. The obtained results show that the thermal conductivity of boron phosphide increases first, and then decreases with increasing pressure, which violates the traditional knowledge that pressure normally enhances thermal conductivity. The explanation for the unusual behavior is believed to be the evolving interplay between AAA and AAO scattering channels with varying pressure. Moreover, it also found that the effect of four-phonon scattering on thermal conductivity is relatively weak at all pressures in boron phosphide. Generally speaking, this paper made a deep investigation towards that the role of selection rules plays in thermal transport. It can be published in this journal. The logical illustration is acceptable. However, there are several problems still needed a clear explanation."

Response: We thank the referee for recommending the publication of this manuscript in Nature Communications. We provide a point-wise explanation for the reviewer's concerns below.

1. The referee asks, "*The atomic configuration of boron phosphide is necessary to be provided, which helps the readers intuitively understand the structure-property relation. How the crystal volumes/structure that directly affects the specific heat in Equ.(2) change with increasing pressure?*"

Response: We have considered the zinc-blende structure of boron phosphide for our current study. The crystal structure is now included in the Supplementary Materials section S1 as Fig. S1. To address the referee's comment, we have revised the main text on pg. 4 to indicate this:

"BP crystallizes in the zinc blende structure and has two atoms in each unit cell. A picture of the crystal structure is given in Fig. S1 of the Supplementary Materials section S1. The phonon dispersions then consist of three A and three O phonon branches."

The measured crystal structure of BP remains stable as zinc-blende up to 110 GPa [41]; our calculated results indicate stability at even higher pressures. To address the reviewer's question about the dependence of the crystal volume on pressure, we have included Fig. R8 (a), which gives the calculated change in volume .vs. pressure, in the Supplementary Materials section S1 (Fig. S2 (a)). We note that our calculated zero-pressure bulk modulus obtained from this data is 170.5 GPa, which is within 1% of the measured value of 172 GPa [44]. To clarify this point, we have added the following text at the end of the Methods section:

The pressure .vs. volume relationship calculated for BP is presented in Fig. S2 (a) of the Supplementary Materials section S1. From it, we obtain a bulk modulus of 170.5 GPa, which is within 1% of the measured value in the literature [44].

Fig. R8 (b) shows the calculated heat capacity of ^{11}BP . Application of hydrostatic pressure decreases the crystal volume, which increases phonon frequencies and decreases phonon occupation numbers. Around room temperature, these effects counteract, giving a roughly constant specific heat with pressure.

We have included Fig. R8 in the Supplementary Materials section S1 (as Fig. S2).

Figure R8: (a) The crystal volume of BP as a function of pressure. The curves for ^{10}BP , ^{11}BP and $^{\text{Nat}}\text{BP}$ were almost overlapping, so only one curve for BP is shown. (b) The volumetric heat capacity of ^{11}BP as a function of pressure at 300 K.

2. The referee says, "*A-O gap is important for competitive relations between 3- and 4- phonon scattering process. I suggest the authors provide a detailed data of A-O gap under the different pressures to clear the relations between A-O gap and different phonon scattering process.*"

Response: We have now included figures that show (a) the absolute magnitude of the A-O gap, and (b) the ratio of the A-O gap to the maximum acoustic phonon frequency, as functions of pressure in ^{10}BP , ^{11}BP , and $^{\text{Nat}}\text{BP}$ in the Supplementary Materials section S3 (Fig. S5). The figures are also reproduced as Fig. R9 below.

With the application of hydrostatic pressure, the A-O gap of BP increases, so the ratio of the A-O gap to the maximum acoustic phonon frequency also increases, thereby forbidding more and more acoustic phonons from participating in the AAO scattering channel. It is precisely this upward shift in the A-O gap that is responsible for uncovering the effect of the AAA selection rule (i.e. the sharp dip in AAA scattering scattering rates) in BP, which is mostly hidden at $P = 0$ GPa. As described in the manuscript, the application of hydrostatic pressure also separates acoustic phonon branches, which gradually turns off the AAA selection rule and increases the AAA scattering rates, *in the region exposed by the vacated AAO processes*. It is this complex interplay between the AAA and AAO scattering channels in BP that is responsible for both the dramatic rise in $k(P)$ and the occurrence of a peak.

Figure R9: (a) The calculated A-O gap, and (b) the ratio of the A-O gap to the maximum acoustic phonon frequency, as a function of pressure in ^{10}BP , ^{11}BP , and $^{\text{Nat}}\text{BP}$ at 300 K.

3. The referee asks, "*Fig. 2 show that the scattering rates of AAA process at 46 GPa have almost one magnitude enhancement than that of 0 GPa. However, Fig. 4 shows that the phonons in range from 0~15 THz contributed thermal conductivity are enhanced under the pressure from 0~34 GPa. In general, the increasing scattering rates will decrease the thermal conductivity. So, what reason could be response for the increasing of thermal conductivity?*"

Response: The referee is highlighting a central point about the competition responsible for the unusual behavior in the pressure-dependent k of BP. Although the AAA scattering rates of BP at ~ 47 GPa are stronger than those at ambient pressure (see Fig. R10(a)), the *total* three-phonon scattering rates, in the mid-acoustic frequency range (5-15 THz) are reduced (see Fig. R10 (b)). This reduction occurs because the competing AAO scattering rates shift to higher phonon frequencies with increasing pressure, due to the influence of the AAO selection rule. This upward shift *exposes*

Figure R10: AAA scattering rates (a) and total three-phonon scattering rates (b) for ^{11}BP at $P = 0$ GPa and $P = 47$ GPa at 300 K.

the dip in AAA scattering rates driven by the AAA selection rule. For pressures below the peak in $k(P)$, there is an sharp reduction in the total scattering rates in the 7-12 THz phonon frequency range.

The reduced scattering rates are responsible for the growing peak in Fig. 5 of the revised manuscript in the $\sim 7\text{-}12$ THz range and contribute to anomalously rapid increase of the $k(P)$ in BP. The low frequency enhancement of k seen in Fig. 5 comes from the increase in acoustic phonon velocities and weakening three-phonon scattering rates from decreased phonon populations with increasing P , which is the behavior typical in most materials.

In the revised submission, we have included Fig. R10 as new Fig. 4 in the main text on pg. 10 (former Fig. 4 has become Fig. 5 in the revised manuscript). We have also added the following additional description to the main text where Fig. 4 is discussed on pgs. 10/11.

The calculated slopes, dk/dP , of the $k(P)$ curves in Fig. 1 around ambient pressure are exceptionally large. Significant contributions to these large values come from the rapid increase in the intrinsic lifetimes of acoustic phonons whose frequencies lie just below the onset of the AAO scattering. This may seem puzzling since the AAA processes are themselves strengthening with increasing P . This is shown in Fig. 4 (a), which plots the AAA scattering rates at $P = 0$ GPa and 47 GPa for $T = 300$ K. The strengthening AAA scattering rates cause the AAA dip to become shallower reflecting the weakening effect of the AAA selection rule caused by the increased separation of the acoustic phonon branches with increasing P . Such increased scattering rates should act to reduce the k of BP. However, the shift of the AAO processes to higher frequencies with increasing P exposes portions of the AAA dip. As a result, the *total* scattering rates actually *weaken* with increasing P in the 7-12 THz range, as shown in Fig. 4 (b). In fact, this feature (i.e., stiffer optic phonons with pressure and their effect on exposing the weak AAA scattering rates) also explains the much larger calculated RT and ambient pressure k of ^{10}BP ($630 \text{ Wm}^{-1}\text{K}^{-1}$) compared with that for ^{11}BP ($530 \text{ Wm}^{-1}\text{K}^{-1}$), consistent with measured results [33, 35].

The reduction in the scattering rates with increasing pressure can be seen in the spectral contributions to k , $k(\nu)$, which are plotted in Fig. 5 for ^{10}BP and ^{11}BP . The reduced total scattering rates seen in Fig. 4 (b) are responsible for growing peaks in Fig. 5 in the $\sim 7\text{-}12$ THz frequency range, and they contribute to anomalously rapid increase of the k of BP with pressure. The effect is enhanced in ^{10}BP compared with ^{11}BP because the lighter ^{10}B mass shifts optic phonons to higher frequencies thereby increasing the A-O gap and exposing more of the AAA dip even at 0 GPa, as seen in Fig. S7 of the Supplementary Materials section S5. The low frequency enhancement of the thermal conductivity seen in Fig. 5 comes from the increase in acoustic phonon velocities and weakening three-phonon scattering rates with increasing P . This is the behavior typical in most materials and leads to the linearly increasing k with P .

4. The referee comments, "*To better understand the results in Fig. 3(a), the effect of temperature on the phonon dispersion should be discussed in detail.*"

Response: Since BP is a strongly-bonded material, its dispersions show only a weak temperature dependence, both at ambient pressure and high pressures, as shown in Fig. R11 (a) and (b) below. Hence, the temperature-dependence of the phonon frequencies themselves do not cause the features observed in Fig. 3 (a) of the manuscript.

The temperature-dependence of the scattering rates, and as a consequence, that of k , as observed in Fig. 3 (a) of the manuscript, comes primarily from the Bose-Einstein distribution functions (Bose factors) in the expressions for the scattering probabilities. As seen from Fig. R11 (c), for any given phonon frequency, the Bose factors sharply increase with respect to T , percentage-wise, at low temperatures, but increase more slowly at high temperatures. This discrepancy of the temperature-dependence of the Bose factors is more prominent for high frequency phonons (optic phonons), than the low frequency phonons (acoustic phonons). Therefore, the temperature-dependence of the Bose factors has a much stronger effect on the AAO scattering channels than the AAA scattering channels due to the participation of one optic phonon mode in the former case. Furthermore, at a fixed temperature, since the phonons stiffen with increasing pressure, their occupation numbers decrease, and this reduction in the phonon occupation happens more sharply at low temperatures.

Due to the above described temperature-dependence of the Bose factors, AAA and AAO scattering rates are comparable in magnitude at 100 K and 0 GPa; however, the AAO scattering rates become much weaker relative to the AAA scattering rates at 100 K as pressure increases. On the other hand, at 300 K and beyond, the total scattering rates are dominated by the AAO scattering channels for the mid-to-high frequency heat carrying phonons even up to a pressure of ~ 50 GPa. More importantly, this trend of the dominance of the AAO scattering channel for mid-to-high frequency phonons remains temperature-independent beyond ~ 300 K. Therefore, at temperatures beyond room temperature, the trends of $k(P)$ relative to $k(P=0)$ remain more-or-less unchanged, as seen in Fig. 3 (a) of the main manuscript.

Figure R11: Phonon dispersions of ^{11}BP at different temperatures at (a) 0 GPa and (b) 47 GPa. (c) The variation of the Bose factor with the phonon frequency at different temperatures. Dashed vertical lines indicate the maximum optic phonon frequency at the displayed pressures for ^{11}BP .

We have modified the main text on pg. 8 to say:

At 100K, the AAO scattering rates simultaneously shift to higher frequencies and weaken rapidly with increasing pressure. However, around and above 300K, their shift to higher frequencies occurs at the same rate as at 100 K, but the weakening of the magnitudes is very minimal. These differences are not caused by changes in phonon dispersions with temperature. Indeed, the phonons in this strongly-bonded compound hardly change over the wide range of T considered in this work (see Figs. S6 (a) and (b) in the Supplementary Materials section S4). Instead, the differences are caused by a more rapid de-population of phonons at 100 K with increasing P compared to that which occurs at higher T, which is quantified by the Bose-Einstein distribution functions plotted in Fig. S6 (c) in the Supplementary Materials section S4.

We have also added expressions for the phonon-phonon scattering probabilities to the Methods section, which explicitly show how the temperature dependence of the probabilities comes from the Bose factors for the different phonons involved in a phonon-phonon scattering process.

5. The referee asks, "*Could the authors evaluate how the electron-phonon interaction affects the obtained thermal conductivity in boron phosphide?*"

Response: First we note that high quality BP samples can now be grown (see Refs. [33, 35] and Ref. [40]), and the intrinsic carrier densities at room temperature and below are very low because of the large energy gap (~ 2 eV), so the residual concentrations of carriers are too low to significantly affect the k of BP (our collaborator, Prof. Bing Lv from the University of Texas at Dallas, who synthesizes the BP samples has told us that the carrier densities are typically well below 10^{18} cm^{-3} in his samples). As evidence for this result, our theoretical calculations of k vs. T for BP have been in good agreement with the measured values for high quality BP samples (see Ref. [33] and Fig. 5 in Ref. [35], also our calculations are reproduced in Fig. R12 below). Since those calculations ignore phonon-electron scattering, it is reasonable to expect that the unusual findings for the $k(P)$ of BP in the present submission should be not be affected by phonon-electron scattering.

Figure R12: Calculated thermal conductivities of ¹¹BP (a) and NatBP (b) with and without four-phonon scattering included, compared with the experimental measurements in Zheng et al. [35]

To address the referee question about the impact of phonon-electron scattering, we have added the following text to the Discussion and Conclusions section:

We note that high quality samples of NatBP and isotopically enriched ¹¹BP and ¹⁰BP can now be fabricated [33, 35]. The intrinsic carrier densities at room temperature and below are very low because of the large energy gap (~2eV). Carrier concentrations measured in some of these samples are less than 10¹⁸ cm⁻³ [40]. Thus, we expect that electron-phonon scattering has minimal influence on the BP thermal conductivity in such samples. In fact, our first principles calculations have shown excellent agreement with the measured thermal conductivities of NatBP and isotopically enriched ¹¹BP and ¹⁰BP without including electron-phonon scattering [33, 35].

6. The referee suggests, "*The authors claimed that the influence of the selection rule on thermal conductivity was long thought to be negligible since intrinsic thermal resistance is dominated by scattering processes involving phonons in different branches, particularly at higher temperatures. Actually, the importance of selection rule has been involved in many previous studies (such as J. Phys.: Condens. Matter 2020, 32, 153002). It is suggested to add a brief discussion on the effect of the breaking of the selection rule on phonon transport and the related reference.*"

Response: When the referee says: "the importance of selection rule has been involved in many previous studies", we presume that the referee is referring to the energy and momentum conservation conditions in a three-phonon scattering process (Eq. 11 of the paper: J. Phys.: Condens. Matter 2020, 32, 153002, now added as Ref. [6] in the main manuscript): $\omega_{\lambda_1}(\mathbf{q}_1) \pm \omega_{\lambda_2}(\mathbf{q}_2) = \omega_{\lambda_3}(\mathbf{q}_3)$ and $\mathbf{q}_1 \pm \mathbf{q}_2 = \mathbf{q}_3 + \mathbf{G}$. In Ref. [6], these conditions are referred to as "selection rules". In other literature, these equations are referred to as general energy and quasi-momentum conservation conditions (or something similar), while the term "selection rules" refers to restrictions on certain specific anharmonic phonon decay channels that follow from the conservation conditions (see e.g. Refs. [3-5, 7-13]). The selection rules that we are discussing (AAA and AAO selection rules) are described in the main manuscript, Supplementary Materials section S2 and Fig. R7 above (now included as part of Supplementary Materials section S2 of the revised manuscript).

We do agree with the referee that the energy conservation conditions have been involved in many previous studies; these conditions impact phonon thermal transport in every material. However, the

effect of the AAA selection rule, as defined herein, is almost never seen in thermal transport because it requires a rare combination of features in a material's phonon dispersions. BAs is perhaps the only example found in nature that demonstrates the large impact of the AAA selection rule through its ultrahigh thermal conductivity at ambient pressure. In BP, the similar impact of the AAA selection rule makes AAA scattering rates very weak, like in BAs, but this effect is hidden by the much stronger AAO scattering rates (see Fig. 2 (b), 0 GPa, main text). Our study shows that it can be exposed through a striking pressure and temperature dependence of its thermal conductivity.

To address the referee's comment, we have added the following to the introduction on pg. 2:

For a given phonon mode, the collection of all such scattering processes that conserve energy and momentum - the scattering phase space - is dictated entirely by a material's phonon dispersions. Thus, for a phonon with wave vector, \mathbf{q} and polarization, j , and phonon frequency, $\omega_j(\mathbf{q})$, the phase space of energy and momentum conserving three-phonon scattering corresponds to all processes satisfying [2-6]: $\omega_j(\mathbf{q}) \pm \omega_j(\mathbf{q}') = \omega_j(\mathbf{q}'')$ and $\mathbf{q} \pm \mathbf{q}' = \mathbf{q}'' + \mathbf{G}$, where \mathbf{G} is a reciprocal lattice vector. Specific features in these dispersions activate selection rules that can severely restrict the phase space of certain phonon-phonon scattering channels [2-5, 7-13].

7. The referee asks, "As mentioned in article [*Phys. Rev. B*, 78 (22), 224303, 2008] and [*Adv. Funct. Mate.*, 30 (8), 1903873, 2020], the lattice thermal conductivity with consideration of full anharmonic effect can also be calculated by atomic Green's function method. What different between AGF method and the theoretical method presented in this work? For examples, whether the contribution of high order Feynman diagram of three- and four- phonon interactions can also be considered in this method?"

The atomistic Green's function (AGF) method is typically used to study transport in nanoscale structures such as nanowires and molecular devices at low temperatures. It is not designed to treat large, macroscopic structures. In contrast, our first principles Boltzmann transport equation (BTE) approach assumes an infinitely extended bulk crystal in three-dimensions, and it exploits the periodic arrangements of the atoms. Our approach can accurately match measurements on bulk samples, and it is predictive since it uses no adjustable parameters. It is not computationally feasible for nanoscale structures, for which AGF approaches are designed. On the other hand, it is computationally challenging to implement first principles based AGF calculations for complex nanostructures that include anharmonicity, so approximate interatomic potential models are used in most such cases. To our knowledge, the higher-order Feynman diagrams have been included within the AGF approach only by using model interatomic potentials i.e. the approach is not first principles.

Our first-principles BTE approach already includes contributions to phonon frequency shifts and lifetimes from the real and imaginary parts of the self energies corresponding to the Feynman diagrams of all three-phonon and the dominant four-phonon interactions, as described in our earlier publications (Refs. [5, 19, 31-33]). It is computationally challenging to include the higher-order Feynman diagrams beyond four-phonon interactions within our first principles framework. To our knowledge, this has not been accomplished to date in the existing literature either.

There are two important reasons why higher-order interactions beyond four-phonon scattering are unlikely to affect the k of strongly-bonded solids like BP.

(1) Using the *ab initio* BTE approach including three-phonon and four-phonon interactions, excellent agreement has been achieved with the measured data for many of these strongly-bonded

materials (see Ref. [5, 19, 33]), so it is unlikely that such higher-order terms are important for the materials considered.

(2) In these strongly-bonded materials, we find that four-phonon scattering affects their k only when the lower-order three-phonon scattering is anomalously weak due to the activation of the three-phonon scattering selection rules. Since we have not found selection rules on four-phonon scattering processes that cause the four-phonon scattering to be anomalously weak (see Ref. [5] for a discussion), we can conclude that higher-order scattering processes beyond those involving four phonons are unlikely to affect the k of strongly-bonded materials.

Response:

Reviewer #3 (Remarks to the Author):

The referee says, "*The authors studied the pressure and temperature dependence of thermal conductivity for BP and found anomalous features including an exceptionally fast rise and a peak with increasing pressure. They stated that the invisible impact of selection rules on three-phonon scattering can be exposed through this kind of anomalous signatures. This manuscript is well organized, clearly written and scientifically sound. The interesting results occur at experimentally accessible conditions and may motivate experimental efforts to observe the predicted novel behavior. I recommend it being published in Nature communications after the following revisions.*"

Response: We thank the referee for recommending the publication of this manuscript in Nature Communications. We provide a point-wise response to the referee's suggestions for revisions, below.

1. The referee asks, "*The highest pressure and temperature that the authors studied is over 100 GPa and up to 1000 K. Does BP keep the the same structure at such high pressure and temperature? The related discussion should be added.*"

Response: BP is a strongly bonded compound that crystallizes in the zinc blende structure at zero pressure. High pressure measurements have found that BP remains stable in the zinc blende structure until at least 110 GPa at room temperature [41]. To our knowledge, no high pressure stability measurements have been performed above room temperature. However, since the calculated phonon dispersions hardly change even at 1000 K (see Fig. R11), we might expect that BP remains stable in the zinc blende structure to high pressures even at 1000 K.

We note that the main findings of the present submission are: (i) the peak and decrease in the BP thermal conductivity, which occurs well below 100 GPa (around 50-60 GPa at 300 K; see Fig. 1 of main manuscript) (ii) the rapid rise in the BP thermal conductivity at low $P \ll 100$ GPa, and (iii) the anomalous decrease in the BP $k(P)/k_0$ as T decreases below 300 K. All of these features occur within the region of stability of BP in the zinc blende structure for which all calculations are performed.

To address this point, we have added the following discussion to the main text in the Discussion and Conclusions section:

Finally, we note that BP remains stable in the zinc blende structure to at least 110 GPa [41]. Thus, all of the anomalous features in the pressure dependent thermal conductivity of BP identified above occur within the region of stability in the zinc blende structure.

2. The referee asks, "It is very interesting to see that the thermal conductivity has a peak as the pressure increases. But the same trend was also predicted for BeSe (PRB 91, 121202(R) 2015) by the same authors, is the reason discussed here in BP suitable for BeSe?"

Response: This is an interesting comparison. In fact, the behaviors in BP and BeSe, though qualitatively similar, arise from different physics and have significantly different magnitudes. The behavior in BeSe is mostly like that in BAs. BeSe is a large mass ratio compound ($M_{\text{Se}}/M_{\text{Be}} \sim 9$), similar to BAs ($M_{\text{As}}/M_{\text{B}} \sim 7$). As a result, BeSe has a large A-O gap and the AAO scattering rates, which are critically important to explaining the anomalous behavior found in BP, do not occur at all in BeSe. As noted in the Fig. 2 caption of the PRB 91 paper noted above by the referee, the peak in the BeSe $k(P)$ curve results from a decrease in the phase space in scattering between an acoustic and two optic phonons (AOO processes). It is a small effect ($\sim 10\%$ increase in the k of BeSe from the ambient conditions to the peak value) that occurs only at low pressure (< 10 GPa). Beyond this pressure, BeSe behaves like BAs i.e. its k limited by 3-phonon scattering decreases with increasing P . In striking contrast, the peak behavior in BP derives from a competition between AAA and AAO three-phonon scattering processes with increasing P , which gives rise to the far more rapid rise in $k(P)$ and the much higher peak value of k , at 250% of its value at ambient pressure.

3. The referee says, "In page 4, 10B in the sentence of (3) B atoms isotopically enriched to 100 % 10B (10BP) should be changed to ^{10}B ."

Response: We thank the referee for pointing out this typo. We have fixed it in the revised submission.

REVIEWERS' COMMENTS

Reviewer #1 (Remarks to the Author):

The author addressed all my concerns and provided an improved version of the paper. They answered all the questions and provided pertinent and well developed explanations for the highlighted points. I enjoyed reading the response letter as much as the manuscript. I do not have new remarks on this version which can be, in my opinion, published in this form.

Reviewer #2 (Remarks to the Author):

The description of the revised manuscript is clear and can be understood better. I would recommend the revised manuscript for publication.

Reviewer #3 (Remarks to the Author):

The authors have addressed all the concerns and incorporate all three referees' suggestions point by point. The response are in detail and reasonable. I suggest that the current version is suitable for publication on Nature Communications, which is also indicated by the other two referees' reports.

Response to referees' comments for the manuscript titled, "Exposing the hidden influence of selection rules on phonon-phonon scattering by pressure and temperature tuning" by Navaneetha K. Ravichandran and David Broido.

Reviewer #1 (Remarks to the Author):

The author addressed all my concerns and provided an improved version of the paper. They answered all the questions and provided pertinent and well developed explanations for the highlighted points. I enjoyed reading the response letter as much as the manuscript. I do not have new remarks on this version which can be, in my opinion, published in this form.

Reviewer #2 (Remarks to the Author):

The description of the revised manuscript is clear and can be understood better. I would recommend the revised manuscript for publication.

Reviewer #3 (Remarks to the Author):

The authors have addressed all the concerns and incorporate all three referees' suggestions point by point. The response are in detail and reasonable. I suggest that the current version is suitable for publication on Nature Communications, which is also indicated by the other two referees' reports.

We thank all three reviewers for taking the time to carefully read our revised manuscript, supplementary materials and response report, and recommending its publication in Nature Communications.